# Differential Expression of NEK Kinase Family Members in Esophageal Adenocarcinoma and Barrett’s Esophagus

**DOI:** 10.3390/cancers15194821

**Published:** 2023-09-30

**Authors:** Lei Chen, Farah Ballout, Heng Lu, Tianling Hu, Shoumin Zhu, Zheng Chen, Dunfa Peng

**Affiliations:** 1Department of Surgery, Miller School of Medicine, University of Miami, Miami, FL 33136, USA; lxc1148@miami.edu (L.C.); fxb414@miami.edu (F.B.); hxl697@miami.edu (H.L.); txh488@miami.edu (T.H.); sxz489@miami.edu (S.Z.); zxc322@miami.edu (Z.C.); 2Sylvester Comprehensive Cancer Center, Miami, FL 33136, USA

**Keywords:** esophageal adenocarcinoma, Barrett’s esophagus, NEK family, gene expression, bioinformatics

## Abstract

**Simple Summary:**

In the present study, we analyzed gene expression of NEK family members (NEK1-NEK11) in esophageal adenocarcinoma (EAC) and its precancerous condition, Barrett’s esophagus (BE), from the TCGA database and eight GEO datasets using a bioinformatics approach. We analyzed genomic alterations of NEKs in EAC samples using cBioPortal For Cancer Genomics and explored the clinical significance of NEKs expression on patients’ survival using the Kaplan–Meier Plotter. We validated the findings using qRT-PCR, Western blotting, immunohistochemistry, and immunofluorescence in cell lines from esophagus and primary EAC tissue samples. Our data indicate that upregulation of several NEKs, such as NEK2, NEK3, and NEK7, may be important in EAC.

**Abstract:**

The incidence of esophageal adenocarcinoma (EAC) has risen rapidly during the past four decades, making it the most common type of esophageal cancer in the USA and Western countries. The NEK (Never in mitosis A (NIMA) related kinase) gene family is a group of serine/threonine kinases with 11 members. Aberrant expression of NEKs has been recently found in a variety of human cancers and plays important roles in tumorigenesis, progression, and drug-resistance. However, the expression of the NEKs in EAC and its precancerous condition (Barrett’s esophagus, BE) has not been investigated. In the present study, we first analyzed the TCGA and 9 GEO databases (a total of 10 databases in which 8 contain EAC and 6 contain BE) using bioinformatic approaches for NEKs expression in EAC and BE. We identified that several NEK members, such as NEK2 (7/8), NEK3 (6/8), and NEK6 (6/8), were significantly upregulated in EAC as compared to normal esophagus samples. Alternatively, NEK1 was downregulated in EAC as compared to the normal esophagus. On the contrary, genomic alterations of these NEKs are not frequent in EAC. We validated the above findings using qRT-PCR and the protein expression of NEKs in EAC cell lines using Western blotting and in primary EAC tissues using immunohistochemistry and immunofluorescence. Our data suggest that frequent upregulation of NEK2, NEK3, and NEK7 may be important in EAC.

## 1. Introduction

Esophageal cancer is one of the major human malignancies, which ranks 7th in incidence and 6th in mortality worldwide [1]. There are two types of esophageal cancers: squamous cell carcinoma (ESCC) and adenocarcinoma (EAC). While the incidence of ESCC has been declining, the incidence of EAC has been increasing alarmingly during the past four decades in the USA and Western countries, making EAC the most common type of malignancy of the esophagus in these countries [2,3,4,5,6]. The estimated age-standardized incidence rate (ASR) for EAC ranges from 1.7 to 3.5 per 100,000 person years in Northern America and most European countries [7,8]. Moreover, the onset age of EAC has become younger [9,10]. The main risk factor for EAC is Barrett’s esophagus [11,12,13,14], a condition that affects the lining of the lower esophagus where the original squamous epithelium is replaced by columnar metaplastic epithelium due to chronic acid reflux in the conditions of gastroesophageal reflux disease (GERD). The estimated population incidence for BE ranges from 0.4% to 5.6% for European ancestral populations [8]. Other risk factors include smoking, obesity, and dietary habits that exclude fresh fruits and vegetables [15,16,17]. However, the prognosis for EAC patients is still poor, with an overall 5-year survival rate of about 20% [6,18]. Although advanced treatment strategies, such as targeted therapy and immune therapy [19,20], have achieved significant progress in some types of cancer, the application of these strategies has lagged in EAC. There is an urgent need to identify and develop novel therapeutic strategies for EAC patients.

The NEK (Never in mitosis A (NIMA) related kinase) gene family is a group of serine/threonine kinases that are related to the NIMA kinase, which was initially found in the fungus Aspergillus nidulans [21,22]. NIMA plays an essential role in regulating mitosis and microtubule dynamics in fungi. The human NEK gene family consists of 11 members (NEK1–NEK11) that share 40–50% of their amino acid sequence identity with NIMA in their catalytic domain [21]. Although NEK family members are relatively newer kinases, accumulating evidence indicates that they play diverse roles in cell cycle regulation, checkpoint control, cilia formation [23], and are associated with several human diseases such as inflammation and cancer [24]. Gene expression and potential functions of NEK members in human cancers are dependent on cancer types. The majority of NEKs, such as NEK1–NEK8, are reported to be upregulated in multiple human cancers and may play an oncogenic role in these tumors. At the same time, NEK9–NEK11 were reported to be downregulated in some tumor types and may have a tumor suppressor function. Though there are publications that analyzed NEKs expression and functions in human cancers [25,26], there is no publication available about NEKs expression in esophageal adenocarcinoma and its precancerous lesions. In the present study, we first used bioinformatic approaches to analyze the gene expression of each NEK family member in esophageal adenocarcinoma from the TCGA database and GEO databases. We analyzed genomic alterations of NEKs in EAC samples and the association between NEKs expression and patients’ survival of EAC. We validated gene expression of NEKs in a panel of esophageal cell lines originating from the normal esophagus, Barrett’s esophagus, dysplastic Barrett’s and esophageal adenocarcinoma, and primary tissue samples of EAC, using quantitative real-time reverse transcription-PCR (qRT-PCR), Western blotting, immunohistochemistry, and immunofluorescence. Our data indicate that some NEKs, such as NEK2, NEK3, and NEK7, are important in EAC.

## 2. Materials and Methods

### 2.1. Analyses of NEKs Gene Expression from TCGA Database and GEO Databases

We conducted screening of gene expression of NEK families throughout public databases, including The Cancer Genome Atlas (TCGA, https://portal.gdc.cancer.gov/repository, accessed on 15 February 2021) and Gene Expression Omnibus (GEO). Specifically, we acquired RNA expression profiles and clinical data pertaining to esophageal adenocarcinomas (EACs) and Barrett’s esophagus. The TCGA dataset encompassed 79 samples of EAC as well as 9 samples of normal esophageal tissue. Additionally, we procured a total of 9 GEO datasets, which include GSE1420 [27], GSE13898 [28], GSE26886 [29], GSE28302 [30], GSE74553 [31], GSE92396 [32], GSE34619 [33], GSE39491 [34], and GSE126304 [35], from the National Center for Biotechnology Information (NCBI) GEO database (https://www.ncbi.nlm.nih.gov/, accessed on 6 March 2021 and 24 August 2023). The expression data were analyzed in the R environment (version 4.2.1). Graphical representations in the form of boxplots were generated using R software. Student’s *t*-test was used to assess differences between two independent variables. Comparisons among multiple groups (≥3 groups) were evaluated through one-way ANOVA, followed by the Bonferroni post hoc test.

### 2.2. Genomic Alterations of NEK Family Members in Esophageal Adenocarcinoma

cBioPortal For Cancer Genomics (https://www.cbioportal.org/, accessed on 20 June 2023) is an open-source resource for interactive exploration of multidimensional cancer genomics datasets [36,37]. We performed genomic analysis using the data from Esophageal Adenocarcinoma (DFCI, Nat Genet 2013) and Esophageal Adenocarcinoma (TCGA, PanCancer Atlas), following the instructions from the website.

### 2.3. Analyses of Association of NEKs Gene Expression and Patients’ Survival

We used Kaplan–Meier Plotter (http://kmplot.com, accessed on 12 June 2023) to analyze the prognostic correlation of NEK expression levels in EAC. Kaplan–Meier Plotter [38] is an online tool to analyze cancer patients’ survival using GEO, EGA, and TCGA databases. Gene expression data and survival information (relapse free and overall survival) are downloaded from GEO, EGA, and TCGA. To analyze the prognostic value of a particular gene, the patient samples are divided into two groups based on quantile expressions assigned for the proposed gene. The Kaplan–Meier survival plot is used to compare the two patient cohorts, and the log rank *p* value is calculated.

### 2.4. Quantitative Real Time RT-PCR

mRNA levels of NEKs were examined from cells of 9 cell lines originating from the esophagus. They are HEEC (normal esophagus), BART and CPA (Barrett’s esophagus), CPB (dysplastic Barrett’s), FLO1, OE19, OE33, SKGT4, and OAC M5.1 (esophageal adenocarcinoma). mRNA levels were further validated in 48 frozen primary human samples that included 18 normal mucosae of the esophagus and 30 samples of esophageal adenocarcinoma. The tissues we used were de-identified coded archival specimens. The Institutional Review Board approved this study as non-human subject research. The RNeasy mini kit (Qiagen, Valencia, CA, USA) was used to isolate and purify the total RNA, and the iScript cDNA synthesis Kit (Bio-Rad, Hercules, CA, USA) was used to synthesize the single-stranded cDNA. We used Primer 3 Plus (Primer3Plus—Pick Primers) online tool to design the primers and acquired the oligos from Integrated DNA Technologies (IDT, Coralville, IA). The sequences of the primers are given in Appendix A. The qRT-PCR was conducted using a Bio-Rad CFX Connect Real Time System (Bio-Rad, Hercules, CA, USA). The threshold cycle number was determined using the CFX Maestro software version 2.0. We triplicated the reactions and averaged the threshold numbers. The results are expressed as relative gene expression fold, normalized to HPRT1, which we tested, showing minimal variations in all normal and tumor samples [39,40].

### 2.5. Western Blotting Analysis

Cell lysates from the above cell lines were prepared in RIPA buffer containing 1× protease cocktail inhibitors (Santa Cruz Biotechnology, Dallas, TX, USA), centrifuged at 12,000 rpm for 15 min at 4 °C. Protein concentrations were measured using the Bio-Rad BSA protein assay (Bio-Rad). Each sample’s protein (15 µg) was loaded into 10% SDS-PAGE gel and transferred onto a nitro-cellulose membrane. After blocking in 5% milk for 1 h, primary antibodies against NEK1 (27146-1-AP, Proteintech, Rosemont, IL, USA), NEK2 (D8, Santa Cruz Biotechnology), NEK3 (12843-1-AP, Proteintech), NEK6 (10378-1-AP, Proteintech), NEK7 (B5, Santa Cruz Biotechnology), and NEK9 (11192-1-AP, Proteintech) were incubated at 4 °C overnight. Secondary antibodies against mouse or rabbit with HRP were incubated for 1 h, then the membrane was developed using Immobilon Western Chemiluminescent HRP Substrate (MilliporeSigma, Burlington, MA, USA) and images were captured using Bio-Rad ChemiDoc XRS+ Imaging system. Band intensity was measured using ImageJ software (version 1.53k) and normalized to β-Actin of the same sample.

### 2.6. Immunohistochemistry

For examination of protein expression of NEKs, a tissue microarray containing 35 EAC and 5 normal esophageal tissues (Each case has two cores) was purchased from TissueArray.com (https://www.tissuearray.com/tissue-arrays/Esophagus/ES8011b, accessed on 26 June 2023). All tissue samples were histologically verified under a microscope. The immunohistochemistry was performed using the Millipore IHC Select HRP/DAB kit (MilliporeSigma) following the manufacturer’s protocol. Briefly, after deparaffinate and rehydration, the slides were boiled in the 10 mM EDTA buffer pH 9 for 15 min for antigen retrieval, followed by 3% H2O2 treatment and blocking using the blocking solution. Antibodies against NEK1 (27146-1-AP, Proteintech) and NEK2 (D8, Santa Cruz Biotechnology) were incubated at 4 °C overnight. After washing in PBS, the slides were incubated with the secondary antibody for 10 min and the Streptavidin HRP for 10 min at room temperature. The slides were incubated with DAB solution and counterstained with hematoxylin. The slides were evaluated under a microscope and the staining intensity and frequency were evaluated as previously described [39].

### 2.7. Immunofluorescence Staining

The same tissue microarray from TissueArray.com ES8011b was used for immunofluorescence staining utilizing antibodies against NEK3 (12843-1-AP, Proteintech) and NEK7 (B5, Santa Cruz). After de-waxing and rehydration by descending concentrations of ethanol, antigen retrieval was performed by boiling slides in 10 mM EDTA, pH 9.0, for 15 min. After blocking in 10% goat serum, the slides were incubated overnight with the above primary antibodies at 4 °C overnight. After washing in PBS, the slides were incubated with Alexa-Flour-488-anti-rabbit or Alexa-Flour-568-anti-mouse secondary antibody for 1 h at room temperature. The slides were covered with VECTASHIELD antifade Mounting Media with DAPI (Vector Laboratories, Newark, CA, USA). Pictures were scanned using the BZ-X710 KEYENCE All-in-one fluorescence microscope (Atlanta, GA, USA) and the fluorescence intensity was quantified using ImageJ software. The mean fluorescence intensity (MFI) per area was adopted and normalized to the MFI of DAPI of the same core.

### 2.8. Statistical Analysis

We used GraphPad Prism software version 9.3.1 for all statistical analyses. The Student’s t test was used to compare the differences between NE and EAC (two groups) if the data were normally distributed, otherwise, the Mann–Whitney test was used. For comparison of multiple groups, the ordinary one-way ANOVA and Dunnett’s multiple comparison tests were applied. For all analyses, *p* < 0.05 was considered as significant.

## 3. Results

### 3.1. Differential Expression of NEK Family Members in Esophageal Adenocarcinoma from TCGA and GEO Databases

To get a comprehensive understanding of gene expression of NEK family members in EAC, we analyzed the gene expression from the TCGA database (Figure 1A) and 7 GEO databases, which contain data from EAC samples (GSE126304 (Figure 1B), GSE26886, GSE13898 (Figure 2), GSE74553 and GSE92396 (Appendix A), and GSE28302 and GSE1420 (Appendix A)). Analyses of gene expression of NEKs revealed differential expression patterns in EAC samples as compared to the normal esophagus. While expression of NEK1 was downregulated in EAC in three of the eight datasets (seven GEO datasets plus the TCGA dataset), expressions of NEK2 (7/8), NEK3 (6/8), NEK6 (6/8), NEK5 (3/8), NEK8 (3/8), NEK9 (2/8), and NEK11 (2/8) were significantly upregulated in EAC, as compared to normal.

Gene expressions of NEK4 and NEK7 were inconsistent among the datasets, showing upregulation in some datasets but downregulation in others or no change. NEK10 is the only NEK that showed no change in all datasets analyzed. Of note, the gene expression of NEK2 is significantly upregulated in EAC in seven of the eight datasets examined. In two early released datasets (Appendix A), NEK2 was the only NEK showing significant upregulation in EAC.

### 3.2. Differential Expression of NEK Family Members in Barrett’s Esophagus from GEO Databases

Because Barrett’s esophagus (BE) is the main known precancerous condition and risk factor of esophageal adenocarcinoma, we next wanted to see the gene expression of each NEK in BE samples as compared to normal esophageal samples. Among the GEO databases we downloaded, there are six databases containing data from BE samples (GSE26886, GSE13898 (Figure 2), GSE28302 and GSE1420 (Appendix A), and GSE34619 and GSE39491 (Appendix A)).

There are big variations in the gene expression of NEKs in BE samples from dataset to dataset. However, upregulation of NEK3 in BE was detected in four of the six datasets, followed by NEK4 (3/6), NEK6 (3/6), NEK9 (3/6), and NEK8 (2/6). These data suggest that alterations of these NEKs are likely to be an early event in Barrett’s tumorigenesis.

### 3.3. Genomic Alterations of NEK Family Members in Esophageal Adenocarcinoma

Genomic alterations, such as gene mutations and copy number variations, are common mechanisms for aberrant gene expression in cancers. To these points, we examined somatic gene mutations and copy number variations of NEK family members through the cBioPortal For Cancer Genomics (https://www.cbioportal.org/, accessed on 20 June 2023). The gene mutation rates (including somatic mutations and copy number alterations) for the 11 NEK families are all below 3% in EAC (Figure 3).

### 3.4. qRT-PCR Validation of NEKs Expression in Human Esophageal Cell Lines and Primary Tumor Tissues

To validate the above bioinformatics analyses results, we applied quantitative real-time RT-PCR (qRT-PCR) analyses using RNAs from esophageal cell lines that originated from normal esophagus (HEEC), Barrett’s esophagus (BAR-T and CPA), and esophageal adenocarcinomas (FLO1, OE19, OE33, SKGT4, and OAC M5.1) (Figure 4). We observed the downregulation of NEK1, NEK9, and NEK11 in cell lines from EAC compared to normal cell lines (Figure 4A,I,K). Whereas the expression of NEK2 and NEK3 are higher in EAC cell lines compared to the normal cell lines (Figure 4B,C). Expressions of NEK4, NEK6, and NEK7 were not consistent among EAC lines, whereas expression levels for NEK5, NEK8, and NEK10 were low in all the cell lines.

To further validate our findings, we performed qRT-PCR from primary EAC tissue samples for NEKs expression (Figure 5). Our data confirmed the downregulation of NEK1 in EAC samples compared to normal esophageal samples and the upregulation of NEK2, NEK3, NEK6, and NEK7 in EAC samples compared to normal samples. No significant alteration was detected for other NEKs, where the expressions of NEK5 and NEK10 were too low to be detected.

### 3.5. NEKs Protein Expression in Human Esophageal Cell Lines and Primary Tumor Tissues

To explore the protein expression of NEKs in EAC, we first examined protein expression levels of NEKs in cell lines that originated from esophageal epithelial cells using a Western blotting analysis (Figure 6). Inconsistent with its mRNA level, we detected high levels of NEK1 protein in several EAC cell lines, compared to the low level NEK1 protein in normal HEEC. Of note, higher protein expression of NEK2 was observed in dysplastic CPB and the EAC cell lines examined, whereas the expression of NEK2 in normal and BE cell lines was relatively low. Only OE19 expresses very high NEK6, whereas other cell lines examined express low levels of NEK6. Dysplastic CPB and SKGT4 express higher levels of NEK7, whereas normal HEEC and BE lines (BART and CPA) express low levels. High levels of NEK9 were observed in most of the cell lines we examined, including BE and EAC.

For further validation, we carried out immunohistochemistry analysis for NEK1 and NEK2 and immunofluorescence staining for NEK3 and NEK7 (antibodies did not work well for IHC) in a tissue microarray containing five normal esophagus and 35 EAC (Figure 7). Normal esophageal epithelia expressed weak (score 1–4) to moderate (score 5–8) levels of NEK1 (Figure 7A), located in both the cytosol and nucleus. However, about half of the EAC (17/35) expressed moderate (score 5–8) to strong (score >= 9) levels of NEK1 in both the cytosol and nucleus (Figure 7B,C). For NEK2, normal esophageal epithelia expressed weak levels of cytosolic staining, while more than half (18/35) of EAC expressed moderate to strong levels of NEK2 in both the cytosol and nucleus. Generally, EAC tissues expressed significantly higher NEK3 (Figure 7G–I) and NEK9 (Figure 7J–L) as compared to normal esophageal epithelial cells, predominantly located in cytosol in both normal and tumor cells for NEK3 and NEK7. These data confirmed the upregulation of these NEKs in EAC in the protein level.

### 3.6. The Significance of NEKs Expression Associated with Patients’ Survival

Gene expression of NEKs has been linked to patients’ survival in human cancers. To explore if the gene expression levels of NEKs in esophageal adenocarcinoma are associated with patients’ survival, we applied an on-line tool of the Kaplan–Meier Plotter (https://kmplot.com/analysis/, accessed on 12 June 2023) to analyze the association of each NEK expression in EAC and patients’ survival. The results are shown in Figure 8. Higher expressions of NEK2 (B), NEK3 (C), and NEK5 (E) are significantly correlated with poorer patients’ survival, while higher expressions of NEK1 (A) and NEK6 (F) are significantly correlated with better patients’ survival. The expression of other NEKs did not reach statistical significance.

## 4. Discussion

Esophageal cancer is one of the major human malignancies, ranking 7th in incidence and 6th in mortality worldwide [1]. In the USA and Western countries, EAC has become the most common type of esophageal malignancy [2,3,4,5,41]. However, the prognosis for EAC patients remains poor, with an overall 5-year survival rate of just about 20%. Unfortunately, there is a big gap between the molecular research on EAC and the clinical needs. There is an urgent need to understand the molecular events underlying the Barrett’s esophagus–dysplasia–EAC cascade. In the present study, we analyzed the gene expression of a relatively new kinase family, the NEK (Never in mitosis A (NIMA) related kinase) gene family of 11 members from the TCGA database and nine GEO datasets. We found differential gene expression patterns of the NEKs in EAC and BE samples. We validated the findings through qRT-PCR, Western blotting, immunohistochemistry, and immunofluorescence staining in esophageal cell lines and primary EAC tissue samples. Our data demonstrated overexpression of some NEKs, such as NEK2, NEK3, and NEK7 in EAC, suggesting their potential roles in EAC.

NEK1 was first identified as a dual specificity kinase with the capacity to phosphorylate serine/threonine and tyrosine, which has about 42% identity with the NIMA in Aspergillus nidulans [42]. The accumulating evidence demonstrated that NEK1 may be involved in the regulation of meiosis [43], DNA damage repair [44,45,46], chromosome stability [47], mitochondrial function [48,49], cell cycle [50], and ciliary function [51,52]. Mutation of NEK1 has been associated with polycystic kidney disease [53] and amyotrophic lateral sclerosis (ALS) [54,55,56]. As for the NEK1 function in cancer, several publications reported overexpression of NEK1 in renal cell carcinoma [57] and gliomas [58]. NEK1 may play an oncogenic function in prostate cancer [59,60] through the phosphorylation of YAP [61]. In our analyses of available databases, we found that gene expression of NEK1 is downregulated in EAC compared to normal samples in TCGA and two GEO datasets (GSE13898 and GSE74553). Our qRT-PCR analyses confirmed that the NEK1 mRNA level is lower in EAC cell lines as compared to normal cell lines (Figure 4). We further demonstrated the downregulation of the NEK1 mRNA level in our primary EAC samples as compared to normal samples (Figure 5).

However, Western blotting analysis of NEK1 protein levels in these cell lines was not in agreement with the mRNA levels; higher levels of NEK1 protein were observed in most of the EAC cell lines compared to the normal cell line. Our IHC analyses displayed higher protein expression levels (scores > 4) in about half of the EAC samples. These data indicate that NEK1 is likely not regulated through transcriptional mechanisms in EAC, but other mechanisms such as through MiRNAs (MiR-136 [62]), translational regulation, and protein stability may be involved. Further functional analyses and molecular signaling pathway investigation are needed to clarify its role in Barrett’s tumorigenesis.

NEK2 is the most closely related mammalian NEK to the serine/threonine protein kinase NIMA of Aspergillus nidulans. A total of 47% sequence identity is found on their catalytic domains and both display a cell cycle-dependent expression pattern with the peak at the G2/M phase [63,64]. Like NEK1, NEK2 has been reported to be involved in the regulation of meiosis [65,66], mitosis and cell cycle [63,64,67,68], and chromosome stability [69]. In addition, NEK2 may also play a role in microtubule stabilization [70], B cell development and immunological response [71], and splicing [72]. NEK2 overexpression has been found in various human malignancies [73,74,75,76,77,78,79,80], including gastrointestinal tumors [81,82]. Aberrant NEK2 expression in cancers has been associated with tumor cell proliferation [83], migration [82], invasion [74,77], drug/radio resistance [84,85], immune response [80,86], and poor prognosis [26,75]. In our bioinformatics analyses, NEK2 is the most frequently upregulated NEK member in EAC (significantly detected in seven of the eight databases). We validated the upregulation of NEK2 mRNA and protein levels in our EAC cell lines and primary EAC tissue samples (Figure 4, Figure 5, Figure 6 and Figure 7). The high NEK2 expression was associated with a poor prognosis in EAC patients (Figure 8). These data strongly indicate that NEK2 plays a crucial role in EAC. Functional and molecular signaling pathway analyses are under investigation in vitro and in vivo.

There are limited publications about the function and roles of NEK3, NEK4, and NEK5 in cancer. NEK3 expression was shown to be downregulated in lung cancer but overexpressed in thyroid carcinoma in one report [87]. Overexpression of NEK3 was associated with poor prognosis in patients with gastric cancer [88], but better prognosis in patients with breast cancer [26]. In our bioinformatics analyses, a significant upregulation of NEK3 was detected in EAC in six of the eight databases, and in BE samples. Our qRT-PCR demonstrated that NEK3 expression was higher in some of EAC cell lines and primary EAC samples than in normal ones. Western blotting showed higher protein levels in BE and EAC lines. Immunofluorescence confirmed a higher expression of NEK3 in EAC compared to normal esophagus. Our data suggest that aberrant expression of NEK3 may play an important role in EAC from an earlier stage of BE. More investigation is needed to clarify the role of NEK3 in Barrett’s tumorigenesis. Available data about NEK4 suggest that it plays a role in the regulation of mitochondrial respiration and morphology [89], DNA damage response, and RNA splicing [90,91]. NEK4 may regulate EMT (epithelial–mesenchymal transition) and promote lung cancer metastasis [92]. In our bioinformatics analyses, the expression of NEK4 in EAC was not consistent among databases. Our qRT-PCR analyses of cell lines and primary EAC tissues did not find significant alteration between EAC and normal samples. Our data suggest that NEK4 may not be involved in EAC. NEK5 may be involved in the regulation of mitosis [93], mitochondrial function [94], and DNA damage response [95]. Overexpression of NEK5 has been reported in thyroid cancer [87] and breast cancer [96,97]. In our bioinformatics analyses, the expression NEK5 in EAC was found to be significantly upregulated in three of the eight databases, suggesting that it may play a role in EAC. However, our qRT-PCR failed to confirm these findings and the level of NEK5 is almost undetectable in our primary tissue samples.

Among NEK family members, NEK6 and NEK7 are the shortest NEKs, which share similar structures, with more than 85% identity on their kinase domain [98]. Although they both can be phosphorylated by NEK9 [99], required for mitotic spindle formation [100], they display distinct expression [101], regulatory [102,103] patterns, and functions [21,104]. Overexpression of NEK6 has been reported in a variety of human cancers [105,106,107,108] and plays an oncogenic function. In our bioinformatics analyses, the overexpression of NEK6 in EAC was frequently detected in six of eight datasets, suggesting its important role in EAC. Our qRT-PCR analysis confirmed that NEK6 expression is significantly upregulated in EAC compared to normal samples. However, we only detected huge protein expression of NEK6 in OE19 cells, while the protein expression levels in other EAC lines are low, like that in normal and BE cell lines. Therefore, the role of NEK6 in EAC is still elusive and needs more investigation. NEK7 has been reported to regulate NLRP3 inflammasome [109,110] and be associated with multiple cancers [111,112,113] and inflammatory diseases [114]. Our qRT-PCR displayed that gene expression of NEK7 is significantly higher in EAC as compared to normal samples, and Western blotting showed high protein levels in dysplastic CPB and some EAC lines as compared to low levels in normal and BE cell lines. Immunofluorescence staining demonstrated a significantly higher expression of NEK7 in primary EAC as compared to normal esophageal epithelia. Considering that Barrett’s related esophageal adenocarcinoma is an inflammation-related disease [115,116], the role of NEK7 in Barrett’s tumorigenesis deserves further investigation.

NEK8 has been reported to be associated with ciliary and centrosome localization and nephronophthisis [117,118]. Recently, it was shown to regulate adipogenesis, glucose homeostasis, and obesity [119]. NEK8 overexpression was found in human breast cancer [120]. However, the data about its role in cancer are limited. Our bioinformatics analyses detected overexpression of NEK8 in three of eight databases. But we failed to confirm the overexpression in our local samples by qRT-PCR. Alterations of gene expression for NEK9-NEK11 were not frequent in our bioinformatics analyses. Our qRT-PCR did not detect significant changes between EAC and normal samples either. Therefore, their roles in EAC tumorigenesis may not be as important as other NEKs.

## 5. Conclusions

Differential gene expressions of NEK family members were detected in EAC and its precancerous Barrett’s esophagus. The overexpression of some NEKs, such as NEK2, NEK3, and NEK7 in EAC, suggests possible important roles in EAC initiation and progression. Additional future mechanistic studies are expected to unravel the molecular and biological functions of these potentially druggable proteins in EAC tumorigenesis.

## Figures and Tables

**Figure 1 cancers-15-04821-f001:**
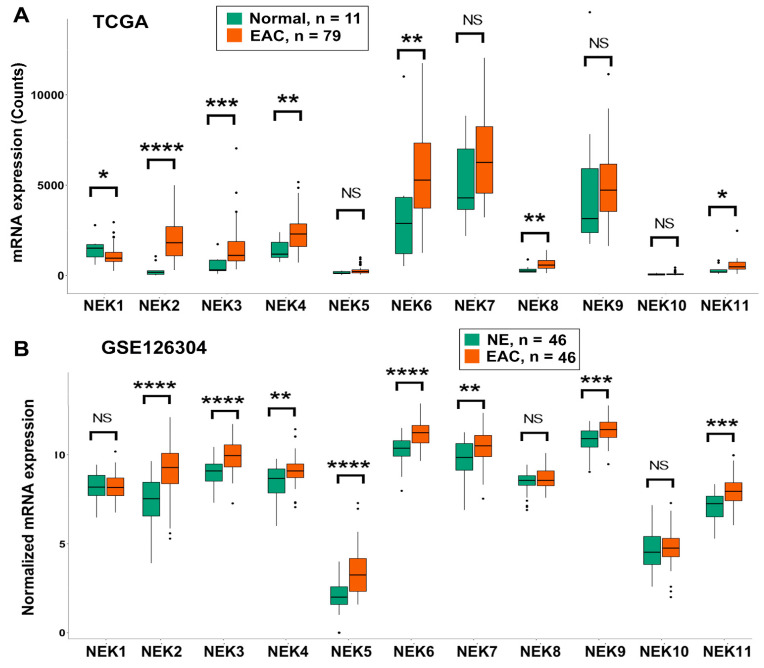
**NEKs gene expression in normal esophagus and esophageal adenocarcinoma**. TCGA dataset (**A**) and GEO dataset GSE126304 (**B**) were analyzed as described in the Methods section. NEKs expressions were presented as the read counts in TCGA dataset (**A**) and the normalized mRNA expression in GSE126304 (**B**) which were based on the original dataset’s normalization in GSE126304. *, *p* < 0.05; **, *p* < 0.01; ***, *p* < 0.001; ****, *p* < 0.0001; NS, not significant.

**Figure 2 cancers-15-04821-f002:**
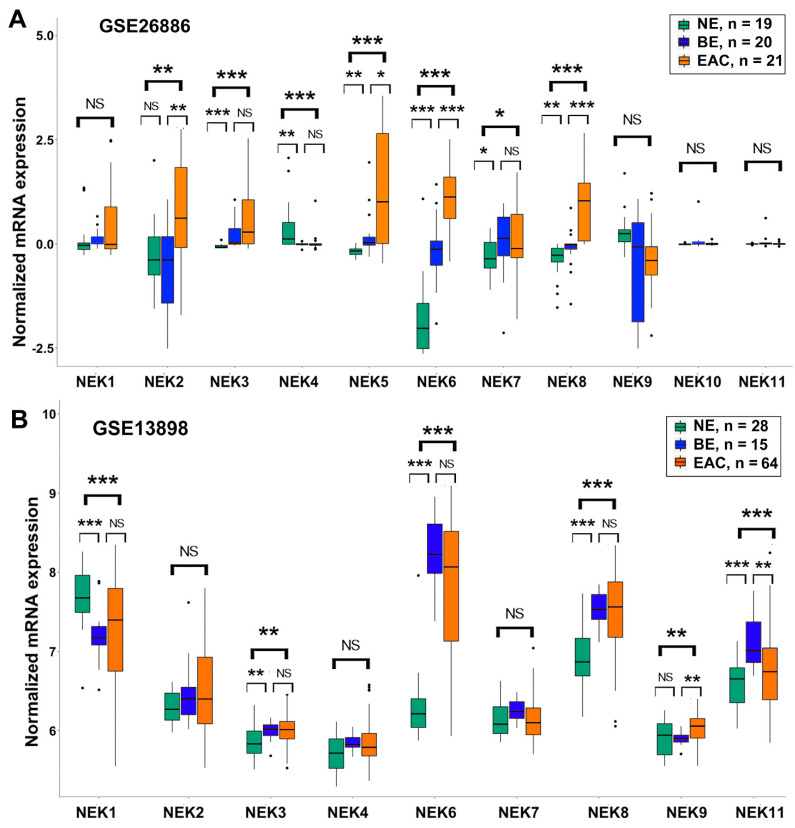
**NEKs gene expression in normal esophagus, Barrett’s esophagus, and esophageal adenocarcinoma**. GEO datasets GSE26886 (**A**) and GSE13898 (**B**), which contain gene expression from normal esophagus, Barrett’s esophagus, and adenocarcinoma samples, were analyzed as described in the Methods section. NEKs expressions were presented as normalized mRNA expression which were based on the original dataset’s normalization. *, *p* < 0.05; **, *p* < 0.01; ***, *p* < 0.001; NS, not significant.

**Figure 3 cancers-15-04821-f003:**
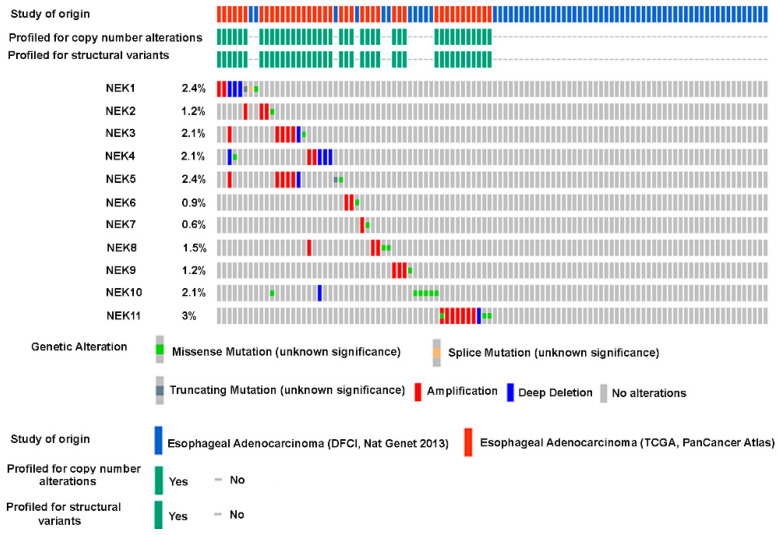
**Genomic alterations of NEKs are rare events in esophageal adenocarcinoma**. An online open-access tool, cBioPortal For Cancer Genomics (https://www.cbioportal.org/, accessed on 20 June 2023), was applied to analyze the genomic alterations that include gene mutations, amplifications, and deletions.

**Figure 4 cancers-15-04821-f004:**
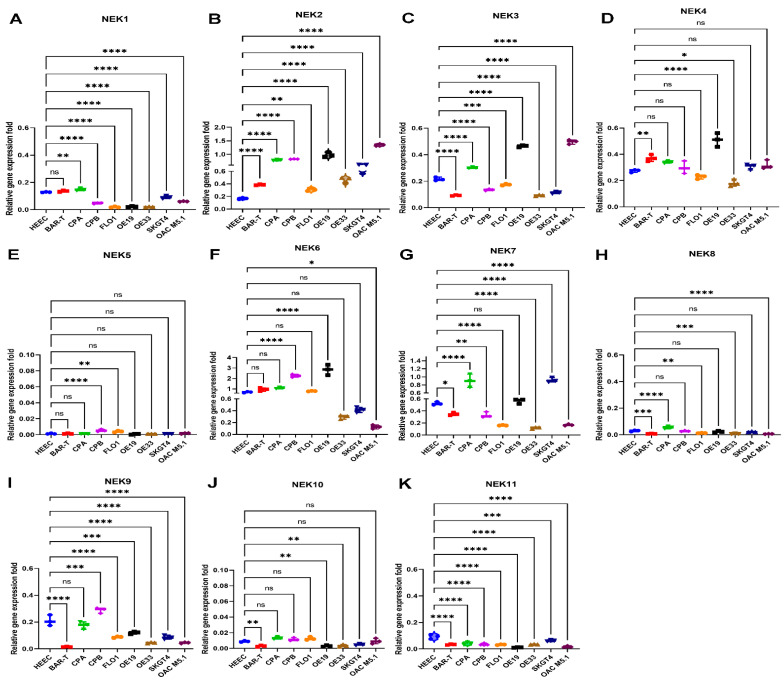
**Quantitative RT-PCR to detect NEKs gene expression in esophageal cell lines.** qRT-PCR was performed as described in the Methods section. The gene expression of NEKs is presented as relative gene expression normalized to the HPRT1 of the same samples. NEK1 (**A**), NEK2 (**B**), NEK3 (**C)**, NEK4 (**D**), NEK5 (**E**), NEK6 (**F**), NEK7 (**G**), NEK8 (**H**), NEK9 (**I**), NEK10 (**J**) and NEK11 (**K**) are shown. HEEC, normal esophagus; BART and CPA, Barrett’s esophagus; CPB, dysplastic Barrett’s; FLO1, OE19, OE33, SKGT4, and OAC M5.1, esophageal adenocarcinoma. Results were compared to the HEEC, a normal control. * *p* < 0.05; ** *p* < 0.01; *** *p* < 0.001; **** *p* < 0.0001; ns, not significant.

**Figure 5 cancers-15-04821-f005:**
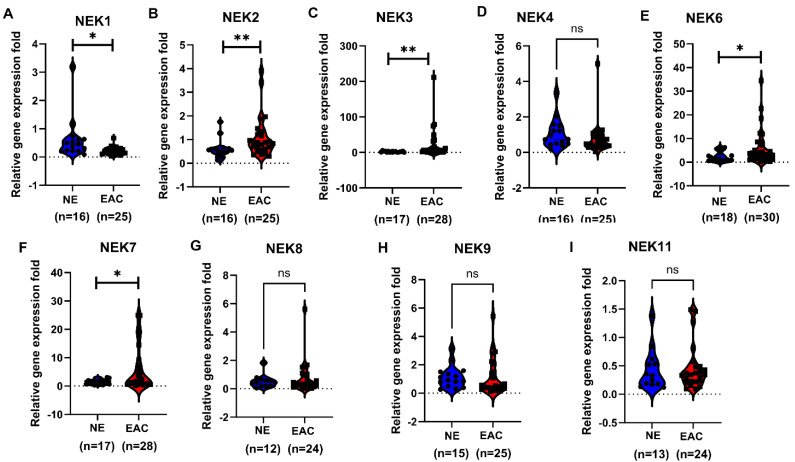
**Quantitative RT-PCR to detect NEKs gene expression in primary tissue samples from normal esophagus (NE) and esophageal adenocarcinoma (EAC).** qRT-PCR was performed as described in the Methods section. NEK1 (**A**), NEK2 (**B**), NEK3 (**C**), NEK4 (**D**), NEK6 (**E**), NEK7 (**F**), NEK8 (**G**), NEK9 (**H**) and NEK11 (I) are presented. The gene expression of NEKs are presented as relative gene expression fold normalized to the HPRT1 of the same samples. *, *p* < 0.05; **, *p* < 0.01; ns, not significant.

**Figure 6 cancers-15-04821-f006:**
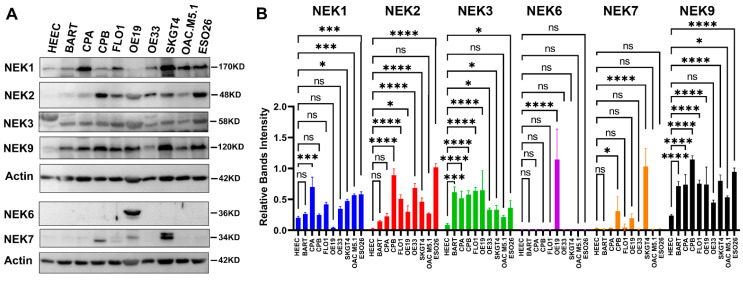
**Western blotting analyses of NEKs protein expression in esophageal cell lines**. Western blotting (**A**) was performed as described in the Methods section (Appendix A). The band intensity was measured using ImageJ software and normalized to the band intensity of Actin of the same sample. The results are shown in (**B**). HEEC, normal esophagus; BART and CPA, Barrett’s esophagus; CPB, dysplastic Barrett’s; FLO1, OE19, OE33, SKGT4, and OAC M5.1, esophageal adenocarcinoma. Results were compared to the HEEC, a normal control. * *p* < 0.05; ** *p* < 0.01; *** *p* < 0.001; **** *p* < 0.0001; ns, not significant.

**Figure 7 cancers-15-04821-f007:**
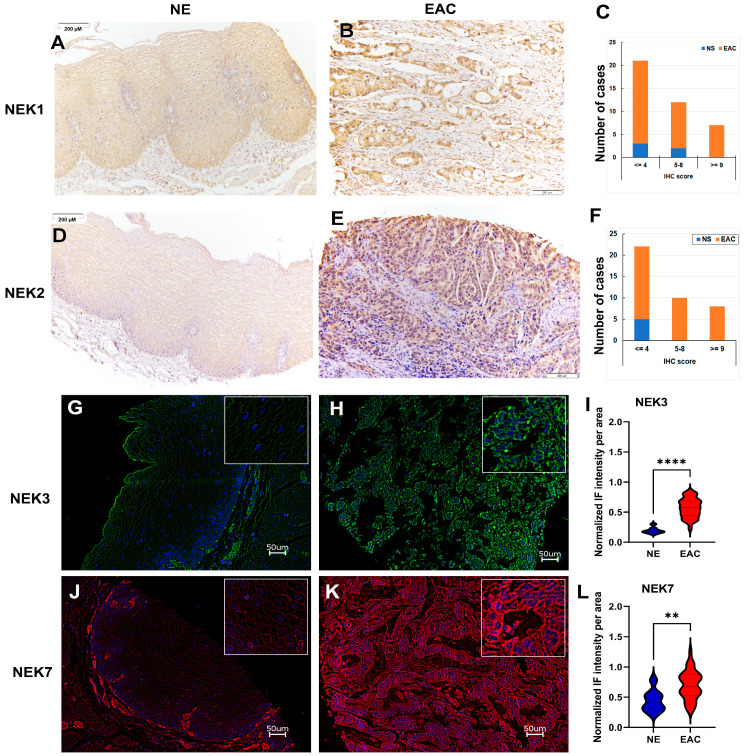
**Immunohistochemistry (IHC) and immunofluorescence (IF) staining of NEKs in a tissue microarray (TMA) containing normal esophagus (NE) and EAC**. (**A**,**B**) IHC of NEK1 in a representative NE (**A**) and EAC (**B**). (**C**) A summary of the IHC results of NEK1 from the TMA. (**D**,**E**) IHC of NEK2 in a representative NE (**D**) and EAC (**E**). (**F**) A summary of the IHC results of NEK2 from the TMA. (G-H) IF staining of NEK3 in a representative NE (**G**) and EAC (**H**). (**I**) A summary of the IF results of NEK3 from the TMA. (J–K) IF staining of NEK7 in a representative NE (**J**) and EAC (**K**). (**L**) A summary of the IF results of NEK7 from the TMA. **, *p* < 0.01; ****, *p* < 0.0001.

**Figure 8 cancers-15-04821-f008:**
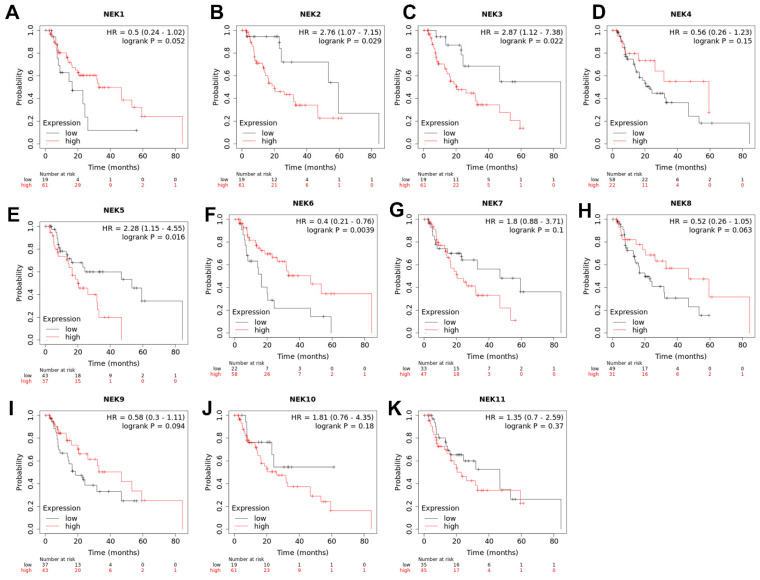
**Analyses of associations of NEKs gene expression and patients’ survival**. An online tool, Kaplan–Meier Plotter (http://kmplot.com, accessed on 12 June 2023), was used to analyze the association between NEKs gene expression levels and EAC patients’ overall survival, according to the instructions of the website. NEK1 (**A**), NEK2 (**B**), NEK3 (**C**), NEK4 (**D**), NEK5 (**E**), NEK6 (**F**), NEK7 (**G**), NEK8 (**H**), NEK9 (**I**), NEK10 (**J**) and NEK11 (**K**) are shown.

## Data Availability

The data supporting this study’s findings are available on request from the corresponding author.

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
