# Peer review of "Differential Expression of NEK Kinase Family Members in Esophageal Adenocarcinoma and Barrett’s Esophagus"

_cancers, 2023, doi:10.3390/cancers15194821_

Round 1
Reviewer 1 Report
Chen and the group have the expression of NEK kinase family members in EAC and BE by studying the TCGA database, GEO datasets, and tissues and cells using RT-PCR, Western blot, and IHC. NEK2, 3, and 6 were found to be upregulated in EAC. Overall, the study is good. I have the following suggestions.
1. Introduction. Please add recent statistics on EAC and BE incidence in Western countries.
2. Figure 4. I see the error bars, but it will be good to see dots as biological replicates to increase the impact of this study. I want authors to compare the mRNA levels of NEK family genes in EAC or BE to normal esophagus lines. So basically, express your data in fold change and use statistics to compare and confirm the upregulation of these genes compared to normal. Similar to what they have done in Figure 5.
3. Figure 6. I see the authors have quantified the western blot. I want to know why only one actin blot has been shown. Have you done all these proteins in one membrane, or was every protein developed with a different membrane, but only representative actin was shown? Moreover, I would like to see the error bars express these values in fold change compared to normal esophageal cell lines and apply statistics to confirm the upregulation.
4. Fig. 7C, F. Please show a dot blot and apply statistics. Similar to Panel I and L.
5. Fig 7. Why were two different methods (IHC and IFC) used to analyze tissues? Add a better picture for NEK1. It looks overstained.
Author Response
We thank the reviewer for the positive evaluation and good suggestions. Please see the attached file for point-to-point responses to your comments.
Thanks again
Best regards
Dunfa Peng

Reviewer 2 Report
The authors analyzed gene expression of NEK family members in esophageal adenocarcinoma (EAC) and its precancerous condition, Barrett’s esophagus (BE) from TCGA database and 8 GEO datasets using bioinformatics approach. They conclude that upregulation of several NEKs, such as NEK2, NEK3 and NEK7 may be important in EAC. Whether from the paper’s organization or the results provided in this study, it is qualified for publishing. Some points should be noticed.
1) For immunohistochemistry, the localization expression of NEKs should be described. Additionally, what are the positive controls for these proteins?
2) As to “These data indicate that the NEK1 is likely not regulated through transcriptional mechanisms in EAC”, please tell more about it, what are the possibilities, e.g. Translation regulation?
3) A recent paper proposes that cancer is a multidimensional spatiotemporal "unity of ecology and evolution" pathological ecosystem. The initiation and progression of human cancer (e.g. the status from the normal to diseased, early to late, local to diffuse) can be considered as an ecological disease (https://www.thno.org/v13p1607.htm). Such new ideas might be helpful for the understanding the progression from the normal esophagus to esophageal adenocarcinoma.
Author Response
We thank the reviewer for the positive evaluation and the good suggestions. Please see the attachment for point-to-point responses to your comments.
Thanks again
Best
Dunfa Peng
